# Comparative Effectiveness and Safety of Concomitant Treatment with Chuna Manual Therapy and Usual Care for Whiplash Injuries: A Multicenter Randomized Controlled Trial

**DOI:** 10.3390/ijerph191710678

**Published:** 2022-08-27

**Authors:** Byung-Jun Kim, A-La Park, Man-Suk Hwang, In Heo, Sun-Young Park, Jae-Heung Cho, Koh-Woon Kim, Jun-Hwan Lee, In-Hyuk Ha, Kyoung-Sun Park, Eui-Hyoung Hwang, Byung-Cheul Shin

**Affiliations:** 1School of Korean Medicine, Pusan National University, Yangsan 50612, Korea; 2Care Policy and Evaluation Centre, Department of Health Policy, London School of Economics and Political Science, London WC2A 2AE, UK; 3Spine & Joint Center, Pusan National University Korean Medicine Hospital, Yangsan 50612, Korea; 4Department of Korean Medicine Rehabilitation, College of Korean Medicine, Kyung Hee University, Seoul 02447, Korea; 5Clinical Medicine Division, Korea Institute of Oriental Medicine, Daejeon 34054, Korea; 6Korean Medicine Life Science, Campus of Korea Institute of Oriental Medicine, University of Science & Technology, Daejeon 34054, Korea; 7Department of Korean Medicine Rehabilitation, Jaseng Hospital of Korean Medicine, Seoul 02453, Korea

**Keywords:** Chuna manual therapy, whiplash injury, multicenter pragmatic randomized controlled trial, comparative effectiveness, Korean medicine

## Abstract

Objectives: We aimed to compare the effectiveness and safety of Chuna manual therapy combined with usual care to those of usual care alone for treating whiplash injuries. Design: A two-arm, parallel, assessor-blinded, multicenter pragmatic randomized clinical trial. Setting: Three hospitals in Korea. Participants: Overall, 132 participants between 19 and 70 years of age, involved in traffic accidents and treated at three hospitals in Korea, >2 but <13 weeks prior to enrollment, with neck pain consistent with whiplash-associated disorder grades I and II and a numeric rating scale score ≥5 were included. Interventions: Participants were equally and randomly allocated to the Chuna manual therapy and usual care (*n* = 66) or usual care (*n* = 66) groups and underwent corresponding treatment for three weeks. Primary and secondary outcome measures: The primary outcome was the number of days to achieve a 50% pain reduction. Secondary outcomes included areas under the 50% numeric rating scale reduction curve: pain, disability, quality of life, and safety. Results: The Chuna manual therapy + usual care group (23.31 ± 21.36 days; *p* = 0.01) required significantly fewer days to achieve 50% pain reduction compared to the usual care group (50.41 ± 48.32 days; *p* = 0.01). Regarding pain severity, functional index, and quality of life index, Chuna manual therapy and usual care were more effective than usual care alone. Safety was acceptable in both groups. Conclusions: In patients with subacute whiplash injury, Chuna manual therapy showed a rapid rate of recovery, high effectiveness, and safety.

## 1. Introduction

The term “whiplash-associated disorder” (WAD) refers to an acceleration-deceleration movement and the resulting energy transfer to the cervical region after an accident [1]. This sudden impact on the cervical spine may cause bone and soft tissue injuries. The primary symptom of WAD is neck pain, and other common symptoms include headache, pain in the shoulder/arm, concentration difficulties, fatigue, and whiplash trauma [2]. WAD grades I and II usually cause soft tissue damage. Patients with WAD grade I or II present with various pain types. However, computed tomography or radiography does not reveal tissue damage. Minimal nociceptive input may arise from slightly damaged tissues in the acute-to-chronic phase [3], where central sensitization could lead to pain and disability in the absence of objective signs of tissue damage in patients with whiplash injury [4].

Treatments of WAD were divided into three groups: acute (less than two weeks), subacute (two to 12 weeks), and chronic (longer than 12 weeks) stages, according to time from injury to a clinician’s assistance in deciding on an appropriate treatment course. Most treatment takes place in the subacute phase for whiplash patients [5] and it is important to early appropriate management is of paramount importance. Inappropriate management at an early stage can cause persistent pain and over-commit resources [6]. Fast management of early-stage pain through active and diverse treatment is necessary.

For WAD patients, South Korea has a dualistic health care system covered by traffic accident insurance that combines Western and Korean medicine. As a Western treatment, medicine and physical therapy are usually provided, and as a Korean medicine treatment, acupuncture, herbal medicine, pharmacopuncture, Chuna manual therapy are usually provided [7]. Korean medicine treatment efficacy has been identified to be more significant than that of analgesics for musculoskeletal disorders [8].

Moreover, one Korean retrospective chart review study found the average number of days for WAD who received Korean medicine treatment was 28.98 days, which was shorter than in others. The mean score for satisfaction in Korean medicine treatment for WAD was 7.16 points, while the most satisfactory treatment among various Korean medicine treatments was Chuna manual therapy (CMT) [9].

CMT is a Korean-style MT practiced by Korean medicine doctors. CMT was primarily derived from traditional Korean MT combined with eastern and western MT [10]. A CMT practitioner employs manual and/or physical force, with the optional use of devices, applying appropriate correcting force to specific areas of the body and restoring balance in orthopedic structure and function [11]. Research on CMT has been steadily increasing and presents the effects on musculoskeletal conditions [12,13]. However, low-quality trials lead to uncertain evidence. So, we conducted well-designed, large-scale study to confirm the effects.

This multicenter pragmatic randomized clinical trial aimed to investigate the following: First, how quickly can CMT reduce neck pain in patients with subacute WAD? Second, if it is effective, how long does the effect last? Third, is it safe? To answer these questions, we compared concomitant CMT with usual care (UC) to UC alone within the comparative effectiveness research framework.

## 2. Methods

### 2.1. Study Design

This two-arm, parallel, assessor-blinded, multicenter pragmatic randomized clinical trial was registered at the Clinical Research Information Service (https://cris.nih.go.kr (accessed on 12 May 2019), KCT0003921) in May 2019. The Institutional Review Boards of the participating institutions approved the study protocol. All participants provided written informed consent. This study was conducted between 20 August 2019, and 2 June 2020, in accordance with the Consolidated Standards of Reporting Trials (CONSORT) guidelines [14], the Committee on Publication Ethics (COPE) guidelines, and the Helsinki Declaration of 1975, as revised in 2013. This paper was prepared based on the thesis of BJK. An economic evaluation reporting the cost per Quality Adjusted Life Years gained will be published separately.

### 2.2. Participants and Settings

The participants were enrolled from three hospitals in Korea (Pusan National University Korean Medicine Hospital, Kyung Hee University Korean Medicine Hospital, and Jaseng Korean Medicine Hospital). The inclusion criteria were as follows: (1) involvement in a traffic accident >2 weeks and <13 weeks prior to enrollment; (2) neck pain consistent with WAD grades I and II; (3) neck pain with a numeric rating scale (NRS) score ≥ 5 (above moderate); (4) age between 19 and 70 years; (5) consent to participate in the trial and provision of written informed consent.

The exclusion criteria were as follows: (1) neck pain resulting from causes other than traffic accident injury; (2) chronic illnesses that may affect the clinical outcomes (cardiovascular disease, kidney disease, diabetic neuropathy, dementia, and epilepsy); (3) absolute contraindications for computerized axial tomography (acute cauda equina syndrome, spinal dislocation, and cerebral aneurysm); (4) previous cervical spine surgery or congenital spinal disorder.

### 2.3. Randomization

The randomization was generated using SAS 9.3 software (SAS Institute Inc., Cary, NC, USA). We used the block randomization method (block size = 4) to assign the same number of participants to each group through a contract research organization. The participant recruitment and group assignment personnel were blinded to the block size. The randomization results were delivered to each center sealed in opaque envelopes. Our design could not allow blinding of the operator and participants; hence, only the assessors were blinded to each site. The evaluators were not involved in the treatment and were blinded to group allocation.

### 2.4. Intervention

#### 2.4.1. CMT + UC Group

After whiplash injury assessment, treatment sessions were applied 2–5 times and 2–3 times in the first and second, and third weeks, respectively. The CMT + UC group was treated by adding only CMT to the treatment of the UC group. CMT was performed by Korean medicine doctors with more than 3 years of clinical experience, and the practitioners underwent pre-study training for standardization. The procedure was divided into two required and three optional CMT techniques to match real clinical conditions. At least one required method and zero or more optional methods were applied in each treatment session. The required CMT methods included the supine position cervical JS distraction and correction. Step 1: push articular process alternately left and right; Step 2: lift articular process alternately left and right; and Step 3: while lifting both sides of the articular process, pull toward the head (Figure 1). Supine position cervical distraction methods: After checking hyperlordosis or hypolordosis, set the angle and segment according to the patient and pull toward to head. (Figure 2). Optional methods included the supine position cervical adjustment, supine atlas adjustment, and/or supine occipital adjustment methods.

#### 2.4.2. UC Group

Patients in the UC group underwent conventional Korean medicine treatments (acupuncture treatment, cupping therapy, physical therapy, herbal medicine, and/or pharmacopuncture), and these were applied for the same number of sessions as the CMT + UC group.

### 2.5. Concomitant Treatments

During the intervention period, there were no restrictions on treatment, including analgesic anti-inflammatory drug administration or physical therapy. However, we monitored their usage to establish between-group comparisons. In the UC group, all treatment combinations, except MT, were allowed and monitored.

### 2.6. Data Collection Procedures

Data were collected from the outpatient departments of the hospitals. The research assistants responsible for data collection were blinded. The NRS was recorded daily by the patients in a separately provided form. Online Appendix A presents the clinical study timeline. Weekly data collection was performed during the 3-week intervention period; further, follow-up (f/u) evaluations were performed at post-intervention weeks 4 (f/u1), 7 (f/u2), 13 (f/u3), and 25 (f/u4) to investigate the short- and mid-term intervention effects.

### 2.7. Outcome Measures

#### 2.7.1. Primary Outcome

The primary outcome was the number of days required to achieve 50% pain reduction (the days taken for the pain NRS score to decrease by 50% from baseline), which was determined using a pain diary provided to participants with instructions to record the daily NRS score for 12 post-registration weeks.

#### 2.7.2. Secondary Outcomes

The secondary outcomes included areas under the 50% NRS reduction curve (NRS-AUC), NRS score, Neck Disability Index (NDI), Patient’s Global Impression of Change (PGIC), a 12-item short-form health survey (SF-12), and adverse events.

An NRS-AUC was used to determine the integral value of the Kaplan-Meier graph for between-group comparisons of the NRS score. The NRS allowed subjective evaluation of neck pain (0, no pain to 10, the worst pain imaginable) [15]. The NDI was developed to assess the degree of neck disability in daily life [16]. The PGIC allows subjective evaluation of symptom improvement in seven steps. Credibility and expectancy have been used to assess patients’ expectations for treatment. The SF-12 is widely used to assess health-related quality of life [17]. Side effects included undesirable and unintended symptoms, signs, and diseases observed at each visit [18].

### 2.8. Statistical Analysis

A previous clinical study that treated a pharmacopuncture treatment group, CMT group, and combination treatment with a CMT group twice a week for 4 weeks was used as a reference to estimate the sample size [19]. The minimum number of participants required for hypothesis testing was calculated using G*Power 3.1.9.2 for Windows (Heinrich-Heine-Universität, Düsseldorf, Germany) as 46 participants per group. Considering a 30% dropout rate, we enrolled 132 participants for the powered sample size. Intention-to-treat analysis was mainly performed concomitantly with per-protocol (PP) analysis. Multiple imputations were performed for missing values. Subgroup analysis was performed by severity.

The sociodemographic characteristics of the participants were evaluated in each group. Continuous variables are expressed as the mean or median. An independent *t*-test was used for between-group comparisons. Categorical variables are expressed as frequencies (%) and were analyzed using the chi-square test and Fisher’s exact test.

Between-group comparisons were performed using an independent *t*-test or analysis of variance (ANOVA). We performed repeated-measures ANOVA to analyze the difference in changes in the trend for each visit. Statistical significance was considered at *p* ≤ 0.05.

## 3. Results

### 3.1. Baseline Characteristics

Among the 138 participants assessed for eligibility, 132 participants were enrolled (50 at Jaseng Korean Medicine Hospital, 44 at Pusan National University Korean Medicine hospital, and 38 at Kyung Hee University Korean Medicine Hospital) and equally allocated to the CMT + UC and UC alone groups. Based on the criteria, we excluded six individuals (Figure 3). There was no between-group difference in the baseline characteristics (Table 1).

### 3.2. Primary Outcome

Days to 50% pain reduction was significantly shorter in the CMT + UC group (23.31 ± 21.36 days) than in the UC group (50.41 ± 48.32 days; *p* = 0.01). Thirty-two patients, including 11 (16.7%) and 21 (31.8%) in the CMT + UC and UC groups, respectively, did not show a decrease of ≥50% in the NRS score by f/u4. The effects of WAD neck pain treatment and the number of patients showing these effects were greater in the CMT + UC group than in the UC group (Table 2).

### 3.3. Secondary Outcomes

Figure 4 shows the survival analysis graph for whiplash-associated disorder neck pain. The NRS-AUC was 479.77 ± 277.10 (median day: 19) and 680.22 ± 297.92 (median day: 53) for the CMT + UC and UC groups, respectively, which showed a significant between-group difference (*p* < 0.001); moreover, the hazard ratio was 2.151.

Table 2 presents the study findings. The results after four weeks of treatment showed considerable improvement. The NRS score significantly decreased in both the CMT + UC (−3.17 ± 1.91) and UC groups (−1.73 ± 1.49) (*p* < 0.001). However, the effects were more significant in the CMT + UC group than in the UC group (*p* < 0.001). The NDI score significantly increased in both the CMT + UC (−9.21 ± 6.49) and UC groups (−6.85 ± 5.73), with the effects being more significant in the CMT + UC group than in the UC group (*p* = 0.028). The PGIC score was better in the CMT + UC group (2.0 [1.0]) than in the UC group (3.0 [1.0]). The SF-12 (physical and mental health composite scores) significantly increased in both the CMT + UC (10.63 ± 8.62, 6.12 ± 8.00) and UC groups (7.90 ± 8.43, 6.14 ± 9.03), with no significant between-group difference (*p* = 0.068, 0.993). An overall analysis of the results revealed that compared to the UC group, the CMT + UC group showed significant pain reduction, except for the NRS score at f/u4. NDI showed improved scores within the f/u period. In SF-12, there were no significant treatment effects.

This study observed 29 adverse events; none of the reactions were serious. Online Appendix A illustrates treatment-related side effects observed in two (3.0%) and three patients (4.5%) in the CMT + UC and UC groups, respectively. The symptoms included headache, fibromyalgia, dizziness, nausea, paranesthesia, and itching.

Online Appendix A illustrates the PP analysis results. Compared with intention-to-treat analysis, PP analysis revealed more significant effects in the CMT + UC group than in the UC group (NRS f/u4, NDI f/u1).

### 3.4. Subgroup Outcomes

The participants were further classified into the WAD1 and WAD2 subgroups for subgroup analysis. Online Appendix A summarizes the demographic and clinical characteristics and results for each subgroup. Despite randomization, there were between-group differences in the baseline characteristics (age, height, sex), which could create bias.

## 4. Discussion

The management of patients with WAD still presents challenges [20]. Over the past few decades, the WAD recovery rates have remained unchanged, with approximately 50% of individuals experiencing active pain and disability [21,22,23,24]. Moreover, the average number of days required for WAD recovery is reported to be approximately 101 days [25]. There has been limited success in managing subacute and chronic WAD [26,27,28,29]. General treatments for WAD involve medication, physical therapy, and other traditional methods for pain relief. However, education, activity, and immobilization are the only evidence-based treatments [30].

A recent study suggests that acute WAD management, prompt return to normal function, pain management, encouragement of self-management, and reductions in fear and anxiety should be considered [31]. Thus, considering the perceived importance of reducing the duration of treatment and the speed of recovery, the primary outcome of this study was identified as the number of days to reach 50% pain reduction. Failure to recover rapidly led to a high percentage of cases with chronic pain and disability and poor quality of life.

CMT has favorable effects on pain and functional improvements caused by musculoskeletal diseases, such as temporomandibular joint disorder, neck pain, lower back pain [32] and safety has also been proven. CMT for patients with WAD involves treating atrophic muscles and stiff spines and joints to promote blood circulation, relieve muscle spasms, remove adherent muscles, and enhance metabolism [33,34]. CMT, one of the non-surgical non-drug treatment methods for WAD, is considered to be an alternative solution for patients who have drug dependence or do not respond to other treatments. The number of studies published on the topic of CMT for traffic accidents has been increasing since 2010, but case-control studies account for a large proportion.

There have been five WAD Korean randomized controlled trials on CMT [35]. Among them, three studies reported that both CMT and UC were effective and useful for WAD; there was no significant difference between the treatment arms. One study indicated that CMT was effective and useful for cervical sprains caused by traffic accidents [36]. Another study reported that the CMT + UC combination therapy was effective for WAD [19]. This is consistent with our finding that CMT is effective for WAD pain alleviation and that the treatment effects are increased in combination therapy.

There are few randomized clinical trials demonstrating the effectiveness of WAD. Although the effectiveness of manual therapy (MT) as a treatment for WAD has been reported and guidelines have been developed accordingly [37], the low quality of the studies and the diverse methods of intervention techniques means that evidence is insufficient to determine the effects of CMT. So, this type of study is required [38]. Comparing the effect size of CMT + UC with other treatment interventions, the NRS change after four weeks of treatment of botulinum toxin is about −2.08 but this study is better at −2.47 [39]. The NRS change after six months of treatment of education + target motor control training is about −2.21 but this study is better at −4.06 [40].

To reflect real clinical conditions, the Korean medicine treatment, which is covered by automobile insurance in South Korea and accounts for approximately half of the medical expenses for automobile accidents, was set as the UC group [41,42]. For the same reason, the treatment type and intensity in the UC group and technique of Chuna in the CMT + UC group were decided by a Korean medicine doctor. Compared to the baseline, the Korean medicine treatment set as the UC group exhibited significant changes in all indicators, including NRS, NDI, and SF-12. Patients in the CMT + UC group showed better NRS and NDI values than those in the UC group. This study involved patients with subacute WAD, considering the likelihood of bias in targeting patients with acute WAD, who may have a faster natural recovery rate regardless of treatment.

In patients with subacute WAD neck pain, concomitant CMT with UC alleviated pain and decreased the days to 50% pain reduction. The NDI was indicative of the positive effects of concomitant CMT. Continuous six-month f/u showed significant positive effects except for some outcomes. NRS-AUC showed a significant difference of 1.4 times, and the hazard ratio was 2.151, suggesting that concomitant CMT therapy was twice as likely to reduce neck pain by at least a half. Moreover, a scientific review reported that although MT reduced neck pain, approximately 50% of patients undergoing MT experienced minor-to-moderate adverse events [43]. In contrast, in our study, 13 patients (9.8%) experienced mildly severe adverse events in the CMT + UC group, of which only two cases were related to treatment.

The strengths of this study lie in its well-designed multicenter clinical study design regarding the treatment of WAD using the Chuna method. Additionally, unlike previous studies that only evaluated the NRS score, this study introduced the 50% pain reduction days from baseline as a temporal index. We conducted a Kaplan-Meier survival analysis and measured quality of life and functional outcomes to comprehensively assess the effects of concomitant CMT. This study has several limitations. First, blinding could not be performed during the intervention period considering the nature of the intervention. Random group allocation and allocation concealment were performed, and the assessors were blinded to prevent selection and detection biases. Second, neither the type nor the intensity of UC was standardized because they were not set in this study, which may have caused bias in the study results. Third, the medical systems for addressing traffic accidents differ across countries. To address this, further collaborative research with other countries is warranted, using conventional drugs and physical therapy for whiplash injuries as a control group.

## 5. Strengths and Limitations of This Study

This study introduced 50% pain reduction days from baseline as a temporal index, in addition to using the pain numerical rating score as an outcome measure.

This was a well-designed multicenter trial with random group allocation and blinding of assessors.

Blinding of participants could not be performed due to the nature of the intervention.

Neither the type nor intensity of usual care was standardized.

## 6. Conclusions

Based on the present study, we found good evidence for CMT showing a rapid recovery rate, high effectiveness, and safety. These results support the need to consider recommending Chuna manual therapies as primary care treatments for WAD neck pain. However, more manual therapy study should be conducted to provide additional evidence.

## Figures and Tables

**Figure 1 ijerph-19-10678-f001:**
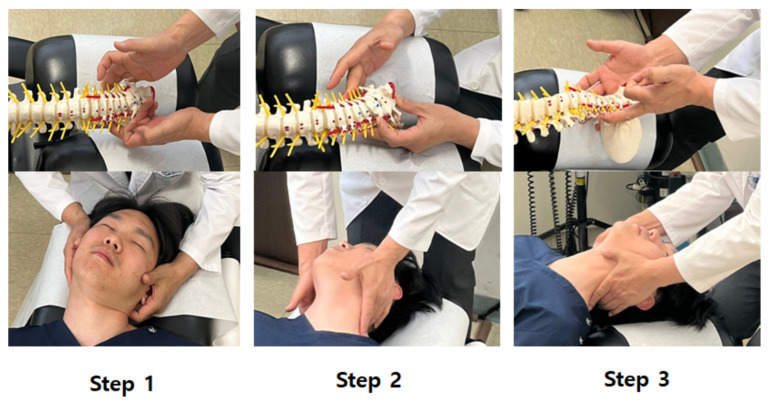
Supine position cervical JS distraction and correction. Step 1, push articular process alternately left and right; Step 2, lift articular process alternately left and right; and Step 3, while lifting both sides of the articular process, traction toward the head.

**Figure 2 ijerph-19-10678-f002:**
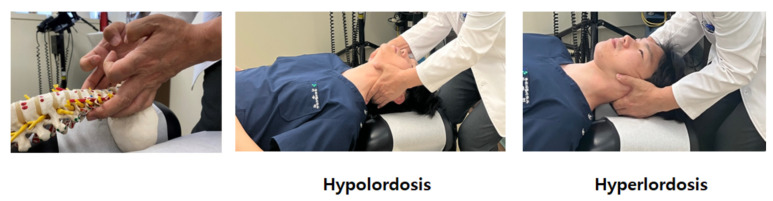
Supine position cervical distraction. After checking hyperlordosis or hypolordosis. Set the angle and segment according to the patient and traction toward to head. For example, IF hyperlordosis, lifting both sides of the C4 articular process and traction 45 degrees toward to head.

**Figure 3 ijerph-19-10678-f003:**
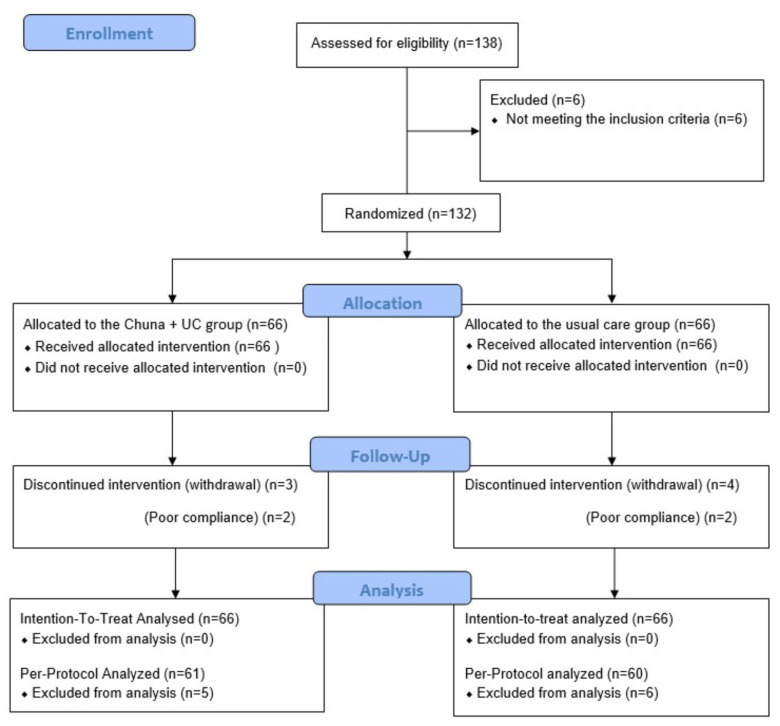
CONSORT flow diagram. Chuna, Chuna manual therapy; UC, usual care.

**Figure 4 ijerph-19-10678-f004:**
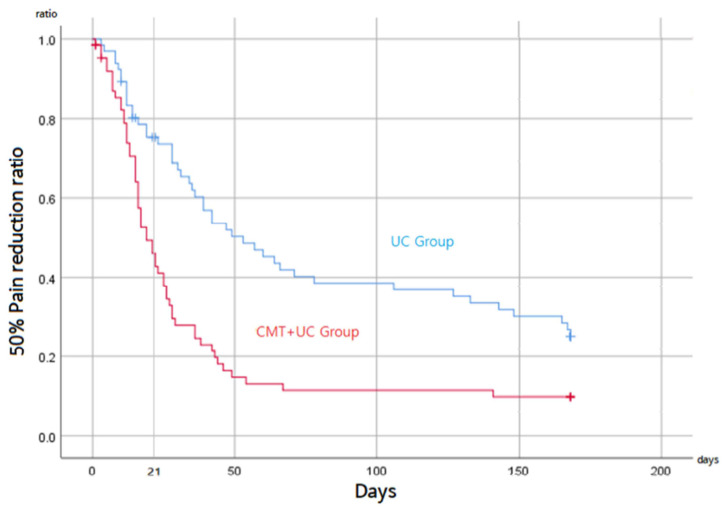
Survival analysis graph for the whiplash-associated disorder neck pain. CMT, Chuna manual treatment; UC, usual care.

**Table 1 ijerph-19-10678-t001:** Baseline characteristics of patients.

Variables		UC Alone (*n* = 66)	CMT + UC (*n* = 66)	*p* Value
Sex *n* (%)	Male	24 (36.4)	17 (25.8)	0.188 ^a^
	Female	42 (63.6)	49 (74.2)	
Age (years)		40.30 ± 12.13	39.61 ± 11.55	0.736 ^b^
BMI		24.12 ± 3.73	22.88 ± 3.56	0.053 ^b^
NRS		6.33 ±1.01	6.33 ± 1.09	1.000 ^b^
NDI		20.24 ± 6.99	19.86 ± 6.85	0.754 ^b^

^a^ *p* values were derived from the chi-square test. ^b^ *p* values were derived from an independent *t*-test for between-group comparisons. BMI, body mass index; CMT, Chuna manual treatment; NDI, neck disability index; NRS, numeric rating scale; UC, usual care.

**Table 2 ijerph-19-10678-t002:** Intention-to-treat analysis of primary and secondary outcomes of whiplash injury treatment.

Variable	Observed Value			Change from Baseline				
	UC Alone (*n* = 66)	CMT + UC (*n* = 66)	*p* Value b	UC Alone (*n* = 66)	*p* Value a	CMT + UC (*n* = 66)	*p* Value a	*p* Value b
**50% pain reduction day**								
F/U 4	50.41 ± 48.32	23.31 ± 21.36	0.001 d 0.001 e	-	-	-	-	-
**NRS_AUC**								
F/U 4	680.22 ± 297.92	479.77 ± 277.10	<0.001 d					
NRS								
visit 2	6.33 ± 1.01 a	6.33 ± 1.09 a	1.000 d					
F/U 1	4.97 ± 1.81 b	3.86 ± 1.92 b	0.001 d	−1.36 ± 1.43	<0.001	−2.47 ± 1.95	<0.001	<0.001 d
F/U 2	4.61 ± 1.82 b	3.17 ± 1.93 c	<0.001 d	−1.73 ± 1.49	<0.001	−3.17 ± 1.91	<0.001	<0.001 d
F/U 3	3.80 ± 2.25 c	2.09 ± 1.84 d	<0.001 d	−2.53 ± 1.77	<0.001	−4.24 ± 2.00	<0.001	<0.001 d
F/U 4	2.98 ± 2.32 d	2.27 ± 2.29 d	0.079 d	−3.35 ± 2.09	<0.001	−4.06 ± 2.42	<0.001	0.072 d
*p* value c	<0.001f	<0.001 f						
NDI								
visit 2	20.24 ± 6.99 a	19.86 ± 6.85 a	0.754 d					
F/U 1	14.98 ± 7.24 b	12.97 ± 6.39 b	0.093 d	−5.26 ± 4.67	<0.001	−6.89 ± 5.53	<0.001	0.068 d
F/U 2	13.39 ± 6.16 b	10.65 ± 5.56 c	0.008 d	−6.85 ± 5.73	<0.001	−9.21 ± 6.49	<0.001	0.028 d
F/U 3	11.64 ± 7.49 c	8.61 ± 5.87 d	0.011 d	−8.61 ± 7.45	<0.001	−11.26 ± 7.10	<0.001	0.038 d
*p* value c	<0.001 f	<0.001 f						
PGIC								
F/U 1	3.0 (1.0) a	2.0 (1.0) a	0.003 e					
F/U 2	3.0 (1.0) a	2.0 (1.0) b	<0.001 e	-	-	-	-	-
F/U 3	2.0 (1.0) b	2.0 (1.0) c	0.001 e	-	-	-	-	-
*p* value c	<0.001 g	<0.001 g						
**Credibility & Expectancy**								
visit 2	7.0 (2.0)	7.0 (3.0)	0.398 e					
**SF-12(PCS)**								
visit 2	39.24 ± 7.10 a	40.10 ± 7.19 a	0.491 d					
F/U 1	43.15 ± 7.43 b	44.78 ± 5.62 b	0.158 d	3.91 ± 7.09	<0.001	4.68 ± 7.09	<0.001	0.534 d
F/U 2	44.12 ± 6.57 b	46.59 ± 6.62 c	0.033 d	4.88 ± 8.24	<0.001	6.49 ± 7.76	<0.001	0.248 d
F/U 3	46.59 ± 7.16 c	48.92 ± 6.73 d	0.057 d	7.35 ± 9.59	<0.001	8.82 ± 8.38	<0.001	0.350 d
*p* value c	<0.001 f	<0.001 f						
**SF-12(MCS)**								
visit 2	38.91 ± 10.92 a	41.03 ± 10.16 a	0.250 d					
F/U 1	45.05 ± 9.41 b	47.16 ± 9.76 b	0.208 d	6.14 ± 9.03	<0.001	6.12 ± 8.00	<0.001	0.993 d
F/U 2	46.81 ± 10.46 bc	51.67 ± 9.31 c	0.006 d	7.90 ± 8.43	<0.001	10.63 ± 8.62	<0.001	0.068 d
F/U 3	49.43 ± 9.61 c	52.22 ± 9.05 c	0.089 d	10.52 ± 9.63	<0.001	11.19 ± 9.39	<0.001	0.688 d
*p* value c	<0.001 f	<0.001 f						

^a^ *p* values were derived from a paired *t*-test for within-group comparisons. ^b^ *p* values were derived from between-group comparisons. ^c^ *p* values were derived from comparing changes over time. ^d^ *p* values were derived from an independent *t*-test. ^e^ *p* values were derived from the Mann-Whitney’s U test. ^f^ *p* values were derived from RM-ANOVA. ^g^ *p* values were derived from the Friedman’s test. AUC, area under the receiver operating characteristic curve; CMT, Chuna manual therapy; f/u, follow-up; f/u1, post-intervention week 4; f/u2, post-intervention week 7; f/u3, post-intervention week 13; f/u4, post-intervention week 25; NDI, neck disability index; NRS, numeric rating scale; PGIC, patient’s global impression of change; RM-ANOVA, repeated measurement analysis of variance; SF-12 (MCS) 12-item short-form health survey mental component summary; SF-12 (PCS), 12-item short-form health survey physical component summary; UC, usual care.

## Data Availability

The data from this study are available upon reasonable request.

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
