# Peer review of "Comparative Effectiveness and Safety of Concomitant Treatment with Chuna Manual Therapy and Usual Care for Whiplash Injuries: A Multicenter Randomized Controlled Trial"

_ijerph, 2022, doi:10.3390/ijerph191710678_

Round 1
Reviewer 1 Report
Dear Authors.
Congratulations for the idea and effort demonstrated.
Attached can see the file with commentaries to enhance and improve the document.
Regards.

Author Response
Authors’ Response to the Review Comments
We appreciate the time given and efforts made by the editor and referees in reviewing this paper. We have addressed all issues indicated in the review report in a point by point manner, and changed those parts in blue. We hope that the revised paper will be able to meet the journal publication requirements.
Once again, we very much appreciate you taking the time to comment on our manuscript.
Response to Comments
(Comment 1) Not enough. REview the notes at the end in Conclusions and rewrite.
Reply) According to the comment, we revised Conclusions
Conclusions: In patients with subacute whiplash injury, Chuna manual therapy showed a rapid rate of recovery, high effectiveness, and safety
(Comment 2) This comments must be in methods not here. -> move to methods
Reply) According to the comment, we move to methods
(Comment 3) Detail the name of Ethics committee, number and date of approval.
Reply) Thank you, we delete this centense
(Comment 4) This is obsolete. Actualize to the 2013 version and cite -
Reply) According to the comment, we revised
(Comment 5) Explain why 70.
Reply) Thank you, we explain why 70
Over 70 years of age were excluded due to Chuna techniques include High-Velocity Low-Amplitude (HVLA) techniques that could occur cervical fracture.
(Comment 6) This is a Key point of the study. Every Manual therapist know about the variability of the manual therapy application, so it must be taken into acount in order to the possible bias it could provoke. It is described the pre-study training for standardizartion, but a complementary statistical analyse shold be recommended by groups, analizing the results from each hospital. So, report this type of statistical analyse in orther to know the normal value of the results.
Reply) The Korean medical doctors completed the common 120 hour chuna basic training course and more than 3 years of clinical experience, and the practitioners underwent pre-study training twice for standardization.
(Comment 7) in general, poor and too short discussion, enlarge. As recommendation, but not only, the effect got with this intervention should be compared with the reported using other therapies (Mckenzie, Maitland...i.e.) and so put in value CMT
Reply) thank you, According to the comment, the discussion revised and the overall content. Duplicate content was deleted and more focus was placed on the effect of CMT.
Fail to rapid recover lead to a high percentage of cases with chronic pain and disability and poor quality of life.
CMT have favourable effects on pain and functional improvements caused by musculoskeletal diseases, such as temporomandibular joint disorder, neck pain, lowback pain [32] and Safety has also been proven. CMT for patients with WAD involves treating atrophic muscles and stiff spines and joints to promote blood circulation, relieve muscle spasms, remove adherent muscles, and enhance metabolism.[33,34] CMT, one of the non-surgical non-drug treatment methods for WAD, is considered to be an alternative solution for patients who have drug dependence or do not respond to other treatments. The number of studies published on the topic of CMT for traffic accidents has been increasing since 2010, but case-control studies account for a large proportion.
There is few randomized clinical trials Studies demonstrated effectiveness treatments for WAD. Although the effectiveness of manual therapy (MT) as a treatment for WAD has been reported and guidelines have been developed accordingly.[38], but the low quality of the studies and the diverse methods of intervention techniques, evidence is insufficient to determine the effects of CMT. So this type of study is required.[39] Comparing the effect size of CMT+UC with other treatment intervention, the NRS change after 4weeks treatment of botulinum toxin is about -2.08 but this study is better at -2.47.[40] the NRS change after 6 month treatment of education + target motor control training is about -2.21 but this study is better at -4.06.[41]
Reviewer 2 Report
First of all, thank you for the opportunity to review your work. It is an extremely interesting work and of great applicability in this type of ailments, especially in people who have suffered a traffic accident.
I would like to make the following considerations:
-Introduction: I think they should go into more depth in this section, addressing concepts related to the research they present in this article.
- The conclusions are very brief, and need to go into more depth in this section. It is an incomplete section.
- Why has it taken so long to release the research results, and can there be new data since June 2020?
Author Response
Authors’ Response to the Review Comments
We appreciate the time given and efforts made by the editor and referees in reviewing this paper. We have addressed all issues indicated in the review report in a point by point manner, and changed those parts in blue. We hope that the revised paper will be able to meet the journal publication requirements.
Once again, we very much appreciate you taking the time to comment on our manuscript.
Response to Comments
(Comment 1) Introduction: I think they should go into more depth in this section, addressing concepts related to the research they present in this article.
Reply) According to the comment, we revised Conclusions
Treatments were divided 3 group, acute (less than two weeks), subacute (two to 12 weeks) and chronic (longer than 12 weeks) stages of WAD according to time from injury to assist clinicians in deciding on an appropriate treatment course.. Most treatment takes place in the subacute phase for whiplash patients.
(Comment 2) The conclusions are very brief, and need to go into more depth in this section. It is an incomplete section.
Reply) According to the comment, we revised Conclusions
Based on present study we found good evidence for the CMT showed a rapid recovery rate, high effectiveness, and safety. In subacute patients approaches seems warranted but more manipulation study should be conducted to provide additional evidence.
(Comment 3) Why has it taken so long to release the research results, and can there be new data since June 2020?
Reply) thank you The reason for the delay in the results of this study was that it was submitted to several journals and it took a long time to obtain the results
Reviewer 3 Report
Dear
I realize that authors have many journals to consider when they want to publish their work, so I appreciate your interest in "Int. J. Environ. Res. Public Health"; I am very sorry not to be able to write in a more positive way. It is evident that you have put a great deal of effort into this project and I want to praise your efforts. Fortunately, the actual contribution from your study is clear and, the manuscript as currently written suggests that it might be suitable for sharing information about this case, but the study that you reported, needs few minor edits. I should like to thank you for give me an opportunity to consider this work for publication. It may be that the you would like to consider resubmitting it, in which case I hope that the comments from my review may help you to revise it before resubmitting it. These comments are given below. Best Regards
- Introduction section: is too poor; more references are missing in the more sentences; I suggest to insert these two papers in introduction section
Mourad F, Patuzzo A, Tenci A, Turcato G, Faletra A, Valdifiori G, Gobbo M, Maselli F, Milano G. Management of whiplash-associated disorder in the Italian emergency department: the feasibility of an evidence-based continuous professional development course provided by physiotherapists. Disabil Rehabil. 2022 May;44(10):2123-2130. doi: 10.1080/09638288.2020.1806936.
Mourad F, Rossettini G, Galeno E, Patuzzo A, Zolla G, Maselli F, Ciolan F, Guerra M, Tosato G, Palese A, Testa M, Ricci G, Zaboli A, Bonora A, Turcato G. Use of Soft Cervical Collar among Whiplash Patients in Two Italian Emergency Departments Is Associated with Persistence of Symptoms: A Propensity Score Matching Analysis. Healthcare (Basel). 2021 Oct 14;9(10). doi: 10.3390/healthcare9101363
- methods: to describe better the Chuna manual therapy and usual care
- in discussion section: Discussions should be reviewed in light of the overall improvement of the paper. Redundant sentences and prewritten information should be avoided. Focus on take-home messages and how that information impacts the clinical practice of management these patients.
- Figures: I suggest must be inserted the figure with Chuna manual therapy example
Author Response
Authors’ Response to the Review Comments
We appreciate the time given and efforts made by the editor and referees in reviewing this paper. We have addressed all issues indicated in the review report in a point by point manner, and changed those parts in blue. We hope that the revised paper will be able to meet the journal publication requirements.
Once again, we very much appreciate you taking the time to comment on our manuscript.
Response to Comments
(Comment 1) introduction section: is too poor; more references are missing in the more sentences; I suggest to insert these two papers in introduction section
Reply) According to the comment, we revised introduction below
Some content have been added, some have been deleted, and the order has also been changed.
Treatments of WAD were divided 3 groups, acute (less than two weeks), subacute (two to 12 weeks) and chronic (longer than 12 weeks) stages, according to time from injury to clinicians's assistance in deciding on an appropriate treatment course. Most treatment takes place in the subacute phase for whiplash patients.[5] and It is important to early appropriate management is of paramount importance. Inappropriate management early stage can cause persist pain and over-commit resources.[6] Fast management of early stage pain through active and diverse treatment is necessary.
For WAD patients South Korea has an dualistic health care system covered by traffic accident insurance that combines Western and Korean medicine. As a Western treatment, medicine and physical therapy are usually provided, and as an Korean medicine treatment, acupuncture, herbal medicine, pharmacopuncture, Chuna manual therapy are usually provided.[7]
Also one korean retrospective chart review study, The average number of days for WAD who received korean medicine treatment was 28.98 days which was shorter than in others. And mean score for satisfaction in korean medicine treatment for WAD was 7.16 points, the most satisfactory treatment among various korean medicine treatment was Chuna manual therapy(CMT). [9]
So we conducted well-desin, large-scale study to confirm the effects.
General treatments for WAD involve medication, physical therapy, and other traditional methods for pain relief. However, education, activity, and immobilization are the only evidence-based treatments.[6] Although the effectiveness of manual therapy (MT) as a treatment for WAD has been reported and guidelines have been developed accordingly.[7,8]
Spinal manual therapy is a common treatment method for neck pain, along with drug and exercise therapy; its safety has been confirmed by Cochrane reviews.[10,11] Spinal manipulative therapy applied to the cervical spine increases hypoalgesia in patients with a concurrent increase in sympathetic nervous system activity.[12,13]
(Comment 2) methods: to describe better the Chuna manual therapy and usual care
Reply) According to the comment, we revised methods below
-> Step 1 push articular process alternately left and right, Step 2 lift articular process alternately left and right, and Step 3 While lifting the both side articular process, pull toward the head.) and supine position cervical distraction methods(After checking hyperlordosis or hypolordosis, set the angle and segment according to the patient and pull toward to head.)
(Comment 3) in discussion section: Discussions should be reviewed in light of the overall improvement of the paper. Redundant sentences and prewritten information should be avoided. Focus on take-home messages and how that information impacts the clinical practice of management these patients.
Reply) thank you, According to the comment, the discussion revised and the overall content. Duplicate content was deleted and more focus was placed on the effect of CMT.
. Fail to rapid recover lead to a high percentage of cases with chronic pain and disability and poor quality of life.
CMT have favourable effects on pain and functional improvements caused by musculoskeletal diseases, such as temporomandibular joint disorder, neck pain, lowback pain [32] and Safety has also been proven. CMT for patients with WAD involves treating atrophic muscles and stiff spines and joints to promote blood circulation, relieve muscle spasms, remove adherent muscles, and enhance metabolism.[33,34] CMT, one of the non-surgical non-drug treatment methods for WAD, is considered to be an alternative solution for patients who have drug dependence or do not respond to other treatments. The number of studies published on the topic of CMT for traffic accidents has been increasing since 2010, but case-control studies account for a large proportion.
There is few randomized clinical trials Studies demonstrated effectiveness treatments for WAD. Although the effectiveness of manual therapy (MT) as a treatment for WAD has been reported and guidelines have been developed accordingly.[38], but the low quality of the studies and the diverse methods of intervention techniques, evidence is insufficient to determine the effects of CMT. So this type of study is required.[39] Comparing the effect size of CMT+UC with other treatment intervention, the NRS change after 4weeks treatment of botulinum toxin is about -2.08 but this study is better at -2.47.[40] the NRS change after 6 month treatment of education + target motor control training is about -2.21 but this study is better at -4.06.[41]
(Comment 4) Figures: I suggest must be inserted the figure with Chuna manual therapy example
Reply) According to the comment we add two figure
|
Figure 2. Supine position cervical distraction
After checking hyperlordosis or hypolordosis, Set the angle and segment according to the patient and traction toward to head. For example IF hyperlordosis, lifting the both side C4 articular process and traction 45 degrees toward to head.
Round 2
Reviewer 2 Report
I believe that after the latest modifications to your article, it meets the standards for publication in this magazine. I would like to take this opportunity to congratulate you on your scientific contribution.